# Determination of Low Muscle Mass by Muscle Surface Index of the First Lumbar Vertebra Using Low-Dose Computed Tomography

**DOI:** 10.3390/jcm11092429

**Published:** 2022-04-26

**Authors:** Ping-Huai Wang, Chien-Hung Gow, Yen-Ling Chiu, Tien-Chi Li

**Affiliations:** 1Division of Pulmonology, Department of Internal Medicine, Far Eastern Memorial Hospital, New Taipei City 220, Taiwan; pinghuaiwang@gmail.com; 2Department of Nursing, Asia Eastern University of Science and Technology, New Taipei City 220, Taiwan; 3Department of Healthcare Information and Management, Ming-Chuan University, Taoyuan 333, Taiwan; 4Graduate Institute of Medicine and Graduate Program in Biomedical Informatics, Yuan Ze University, Taoyuan 320, Taiwan; yenling.chiu@gmail.com; 5Division of Nephrology, Department of Internal Medicine, Far Eastern Memorial Hospital, New Taipei City 220, Taiwan; 6Graduate Institute of Clinical Medicine, National Taiwan University, Taipei 100, Taiwan; 7Department of Radiology, Far Eastern Memorial Hospital, New Taipei City 220, Taiwan; guardian0331@gmail.com

**Keywords:** computed tomography, muscle surface index, low muscle mass, low-dose CT, sarcopenia

## Abstract

The muscle index of the first vertebra (L1MI) derived from computed tomography (CT) is an indicator of total skeletal muscle mass. Nevertheless, the cutoff value and utility of L1MI derived from low-dose chest CT (LDCT) remain unclear. Adults who received LDCT for health check-ups in 2017 were enrolled. The cutoff values of L1MI were established in subjects aged 20–60 years. The cutoff values were used in chronic obstructive pulmonary disease (COPD) patients to determine muscle quantity. A total of 1780 healthy subjects were enrolled. Subjects (*n* = 1393) aged 20–60 years were defined as the reference group. The sex-specific cutoff values of L1MI were 26.2 cm^2^/m^2^ for males and 20.9 cm^2^/m^2^ for females. Six subjects in the COPD group (6/44, 13.6%) had low L1MI. COPD subjects with low L1MI had lower forced expiratory volume in one second (0.81 ± 0.17 vs. 1.30 ± 0.55 L/s, *p* = 0.046) and higher COPD assessment test scores (19.5 ± 2.6 vs. 15.0 ± 4.9, *p* = 0.015) than those with normal L1MI. In conclusion, LDCT in health assessments may provide additional information on sarcopenia.

## 1. Introduction

Loss of skeletal muscle mass commonly occurs in older adults, and is associated with poor clinical outcomes [1]. Sarcopenia is an important geriatric syndrome characterized by muscle loss and dysfunction, and it worsens with aging [2]. Clinically, skeletal muscle loss is associated with significant health problems, including functional impairment, disability, risk of fractures, falls, and increased length of hospital stay [3,4,5]. Additionally, sarcopenia contributes to nosocomial infections and decreased survival in non-cancer diseases [6,7]. Therefore, early diagnosis of skeletal muscle loss and provision of adequate interventions are warranted to improve clinical outcomes [8].

Quantitative analysis of skeletal muscle mass is a key component of sarcopenia diagnosis [9]. Clinically, CT-measured skeletal muscle mass is strongly correlated to sarcopenia [9,10,11]. The European Working Group on Sarcopenia in Older People (EWGSOP) proposed and qualified CT as one of the gold standard imaging modalities for measuring muscle mass [12]. Previous studies assessed a single-slice abdominal CT image and demonstrated that the cross-sectional muscle area of the third lumbar vertebra (L3) correlated well with total body muscle mass in healthy adults [13,14,15]. Furthermore, the EWGSOP report in 2019 recommended the cross-sectional muscle area of the L3 level, corrected by height squared (L3MI), as a parameter for diagnosing low muscle mass [16]. However, clinical application of conventional CT in this manner, especially in the healthy population, is limited by accessibility concerns due to the indications and radiation of CT exams [17].

A strong correlation between the cross section of the muscle area at the first lumbar vertebra (L1) and the L3 level has been suggested [18,19,20]. This relation implies that the cross-sectional muscle area at L1 level could be an alternative to that at L3 for evaluating the status of skeletal muscle mass [18,19,20]. The correlation makes it feasible that muscle mass status could be evaluated from the L1 level with a chest CT, not only abdominal CT. Recently, low-dose chest CT (LDCT) has been widely adopted in healthy populations for lung cancer screening [21,22]. Additionally, LDCT provides accurate and reproducible measurement of a cross-sectional muscle area as compared to conventional CT or magnetic resonance imaging (MRI) [23,24,25]. Therefore, it could be plausible that the cross-sectional muscle area at the L1 level derived from LDCT is used to evaluate the status of skeletal muscle mass, without being limited to patients with specific diseases. The diagnostic criteria of the cross-sectional muscle area at the L1 level, corrected by height squared (L1MI), are essentially required for identification of low muscle mass. However, little information is available on the cutoff values of L1MI. Therefore, this study aimed to determine the cutoff values of L1MI derived from LDCT in the young reference group. As mentioned above, there are no universal gold standard of L1MI to determine low muscle mass. However, it is well known about the relationship of sarcopenia and COPD severity [26,27]. Therefore, we further applied the L1MI diagnostic criteria to COPD patients with chest CT and investigated the correlation of the status of skeletal muscle mass and disease severity to prove the utility of the cut-off points.

## 2. Materials and Methods

### 2.1. Participants

We retrospectively enrolled adults aged 20 years and older who received LDCT in routine health check-ups at a tertiary care hospital from January to December 2017. Subjects completed medical history questionnaires before commencing the health check-ups. LDCT images, demographic information, medical history, and anthropometric data, including fat-free muscle mass (FFM) measured by bioelectric impedance analysis (BIA) (Appendix B), were collected. Participants with major organ dysfunction (Appendix C) or malignancy were excluded. Subjects aged 20 to 60 years were grouped as the reference group to develop the cutoff points of L1MI [28]. Subjects over 60 years of age were categorized as the older group [29]. Meanwhile, patients who enrolled in COPD case management (Appendix D) at the hospital were screened from July to September 2017. We retrieved clinical information for COPD patients who received chest CT examinations in 2017 as the COPD group. This study was approved by the Research Ethics Review Committee of the Far Eastern Memorial Hospital (IRB-107091-E and IRB-109098-E). All the personal data were delinked, and informed consent was waived due to the retrospective nature of the study.

### 2.2. LDCT Image Acquisition and LDCT-Based Determination of Low Skeletal Muscle Mass

LDCT was performed using a dual-source 128-slice CT scanner (Somatom Definition Flash, Siemens Healthcare, Forchheim, Germany). Scanning was performed from the thoracic inlet to the middle portion of the kidneys using the following scanning parameters: tube voltage, 120 kVp; collimation of 128 × 0.6 mm; scanning range, 35 cm; pitch, 0.75; rotation time, 0.33 s; kernel (convolution algorithm, image reconstruction for CT), I26 for soft tissue window and B70 for lung window. Lumbar skeletal muscle index was determined by LDCT-based cross-sectional imaging. The L1 level was identified as the level of the inferior endplate of the L1 vertebra. The target images were selected by two pulmonologists independently, which were confirmed by the radiologist. Body composition was segmented by SliceOmatic v5.0 software (TomoVision, Montreal, QC, Canada). Tissue-specific Hounsfield unit (HU) thresholds (−29–150 HU) were used to highlight muscle areas (Appendix A). SliceOmatic software was used to calculate the muscle area. L1MI (cm^2^/m^2^) denoted that the cross-sectional muscle surface area at the L1 level was normalized to stature by division by height squared.

### 2.3. Development of Cut-Off Values of L1MI

The normal references for L1MI were derived from the reference group. The sex-specific cutoff values of low L1MI were defined as the values of two standard deviations (SD) below the mean [12]. According to the cutoff values obtained from the reference group, we determined the status of L1MI in the older subjects.

### 2.4. The Use of the Sex Specific L1MI Cutoff Values in COPD Patients

The chest CT of COPD patients was analyzed in the same way as the LDCT. The data of post-bronchodilator spirometry tests in 2017 were recorded. Meanwhile, the body weight, height, handgrip strength (HGS) (Appendix E), and COPD assessment tests (CAT) [30] within 3 months before or after CT exams were collected. In cases of multiple data of the aforementioned items, the data closest to the dates of CT exams were chosen. The definition of exacerbation was aggravation of respiratory symptoms and respiratory distress which required oral steroids and antibiotics, emergency visits, or hospital admission. The day of the chest CT examination was denoted as the index day. Frequent exacerbation in the previous year was defined as an exacerbation history of the subject visiting the emergency room more than once or having been admitted, within one year before the index day.

### 2.5. Statistical Analyses

The Chi-squared test was used to compare categorical variables between men and women. Continuous variables were compared using the Student’s *t*-test or Mann–Whitney U test. Pearson’s correlation analysis was used to assess relationships between L1MI by LDCT and continuous variables. All analyses were performed in SPSS software (version 19.0 for Windows; SPSS Inc., Chicago, IL, USA)

## 3. Results

### 3.1. Characteristics of Subjects and Determination of Sex-Specific L1MI Cutoffs

A total of 1780 subjects who received LDCT examinations were included in this study. Their characteristics are summarized in Table 1. The mean age was 51.2 ± 11.1 years old. The sample comprised 1129 males (63.4%) and 651 females (36.6%). No sex-specific age differences were noted.

For most of the subjects (88.6% of the males and 88.7% of the females), body mass index (BMI) ranged between 18.5 and 30. Height and weight were greater in males than in females (170.6 ± 6.3 vs. 158.3 ± 5.9 cm, *p* < 0.001 and 74.3 ± 11.5 vs. 58.3 ± 10.0 kg, *p* < 0.001). The skeletal muscle index (SMI) was defined as FFM (kg)/height squared (m^2^). As expected, males had higher SMI and L1MI than females (17.3 ± 5.9 vs. 13.8 ± 4.7 kg/m^2^; 38.4 ± 6.1 vs. 29.7 ± 4.4 cm^2^/m^2^, both *p* < 0.001). The Pearson correlation coefficient r of SMI and L1MI was 0.251, *p* < 0.001. There was no significant difference in L1MI among different ages by decade (Figure 1). The sex-specific cutoff points of L1MI were 26.2 cm^2^/m^2^ for males and 20.9 cm^2^/m^2^ for females.

### 3.2. Comparison between the Reference and Older Groups

The older group consisted of 255 males and 132 females. The characteristics of this group are listed in Table 2. The mean ages of the males and females were 66.0 ± 4.5 and 65.9 ± 4.9 years, respectively. Compared to those of males in the reference group, the height and weight of the males of the older group were lower (167.2 ± 5.7 vs. 171.8 ± 6.3 cm, *p* < 0.001; 70.8 ± 9.6 vs. 75.5 ± 11.7 kg, *p* < 0.001). No significant difference in BMI was observed. In comparison to females of the reference group, height was lower in females of the older group (154.4 ± 5.6 vs. 158.8 ± 5.3 cm, *p* < 0.001) but body weight was similar. Thereafter, the female BMI of the older group was higher than that of the reference group (24.5 ± 4.0 vs. 23.1 ± 3.7 kg/m^2^, *p* = 0.001). Regardless of sex, L1MI of the older group was similar to that of the reference group. Based on the diagnostic criteria of low L1MI developed by the reference group, twelve males (12/255, 4.7%) and none of the females were categorized as low L1MI (Figure 2).

### 3.3. Application of the Diagnostic Criteria for Low L1MI in COPD Patients

We screened 273 COPD patients and enrolled 44 patients with chest CT scans for L1MI measurement. Six patients (6/44, 13.6%) were determined as low L1MI, based on the cut-off points of L1MI. The clinical characteristics of COPD patients with low and normal L1MI are presented in Table 3. No significant differences were observed in age, sex, comorbidities, or height between low L1MI and normal L1MI groups. The Pearson correlation coefficient of L1MI and SMI was 0.682, *p* < 0.001. However, weight and BMI were significantly lower in the low L1MI than in the normal L1MI group (49.1 ± 6.2 vs. 66.1 ± 11.7 kg, *p* < 0.001; 19.0 ± 2.5 vs. 25.0 ± 4.2 kg/m^2^, *p* < 0.001). COPD subjects with low L1MI had significantly higher CAT scores (19.5 ± 2.6 vs. 15.0 ± 4.9, *p* = 0.015), lower forced expiratory volume in one second (FEV_1_) (0.81 ± 0.17 vs. 1.30 ± 0.55 L/s, *p* = 0.046), and marginally lower HGS (19.5 ± 2.6 vs. 24.4 ± 6.7 kg, *p* = 0.098), compared to the normal L1MI group (Table 3 and Figure 3). Subjects with low L1MI also tended to have lower FEV_1_% of the predicted value (40.1 ± 11.8% vs. 56.5 ± 20.1%, *p* = 0.113). Although the occurrences of frequent exacerbation in the past year were higher in the low L1MI group than in the normal L1MI group, the difference was not significant [50% (3/6) vs. 23.7% (9/38), *p* = 0.321].

## 4. Discussion

In the present study, we utilized L1MI data derived from LDCT in a normal younger population to determine the sex-specific criteria of low L1MI. The sex-specific cutoff values of L1MI were found to be 26.2 cm^2^/m^2^ for males and 20.9 cm^2^/m^2^ for females. We also reported that the diagnostic criteria of L1MI were used in COPD patients. Compared to normal L1MI, COPD patients with low L1MI had worse clinical symptoms, poor lung function, and possibly lower HGS, which echoed the findings of previous studies on the relationship of sarcopenia and COPD severity [26,27].

L3MI was recommended by EWGSOP guidelines in 2019 as a diagnostic modality of low muscle mass [16]. Initially, this method was proposed for evaluation of muscle mass status in cancer populations [15,31,32]. Apart from the information that CT provides on cancer status, L3MI derived from CT yields additional information about the status of skeletal muscle mass. However, the application of L3MI in the general population has limitations because abdominal CT is not routinely employed in it [14,20]. Previous studies proposed L1 as an alternative level to L3 for the evaluation of cross-sectional surface areas of muscles [18,19]. Therefore, utilizing L1MI is more applicable in clinical settings because both abdominal and chest CT provide measurable muscle surfaces at the L1 level.

EWGSOP recommended that the cutoff value of one specific diagnostic modality for low skeletal muscle mass was two SDs below the mean in a normal young population [12]. However, certain difficulties impede the development of criteria on the cross-sectional muscle area measured by CT for a diagnosis of low skeletal muscle mass. The main difficulty is that CT is usually used in patients with specific diseases and rarely in healthy individuals. Therefore, the data for the L3MI criteria were mainly derived from the studies of healthy donors of organs for transplantation [14,28] or explorations of other measurement criteria, such as BIA or dual energy X-ray absorptiometry in ill populations [15]. The majority of these studies had limited case numbers, which might have affected the predictive power. Recently, a large-scale study by Kim and colleagues tried to overcome the problem and reported diagnostic criteria of low L3MI derived from abdominal CT data collected in health check-ups [33]. The aforementioned difficulty in establishing L3MI diagnostic criteria also exists in obtaining those of L1MI [18,20,34]. The strength of our study is that it was a large cohort study analyzing data from more than 1000 healthy young subjects. Furthermore, the application of LDCT in routine health assessment further resolved the problem that chest or abdominal CT is not usually applied to a normal healthy population. Furthermore, follow-up of LDCT at regular intervals for lung cancer screening provided additional longitudinal details of muscle changes.

Various cutoff values of L1MI for different populations have been proposed in previous studies. Those values have ranges of 34.5–52.4 cm^2^/m^2^ for males and 38.5–26 cm^2^/m^2^ for females [18,20,33]. The diagnostic criteria of low L1MI proposed by the present study are slightly different from the above data. The L1MI cutoff values of some studies were derived indirectly from extrapolation of L3MI data or clinical outcomes of specific diseases [18,34]. The data of L1MI reported by Derstine and his colleagues, which was derived from the CT exams of renal donors, is relatively close to our result [20]. The differences in the two studies might have been contributed by ethnicity or regional and generation effects, as the subjects were in the US and the period of enrollment distributed over more than 10 years [35]. The cutoff value of the present study should be more applicable in our region at least because we demonstrated that it could differentiate the disease severity of COPD.

Previous studies have reported a 20% decline in cross-sectional muscle areas in aged populations [36]. However, our data revealed that the means of L1MI values among ages by decades did not differ significantly. Moreover, no female subjects over 60 years of age were categorized as having low L1MI status. Wen and his colleague also reported a similar finding in China; none of the older subjects were diagnosed with low muscle mass based on criteria derived from younger ones [37]. This similarity might be associated with the daily physical activity and lifestyle shifts in different Asian generations. The effect of age on L1MI may be counteracted by the higher past physical activity of the older adults, which could be related to past physical labor in individuals with a history of low economic status, while L1MI may be reduced by the western lifestyle and convenient transportation in the current younger population [38]. In addition, our older subjects were reported to have fair socioeconomic and health status. Therefore, the muscle mass of older subjects could be maintained at the level of that of the younger generation [29].

Most studies define the ages of 20 to 40 years as the young reference group for the cutoff value of diagnosing low muscle mass [39]. In this study, we adopted age span of 20 to 60 years as the young reference group. The reasons for our reference group are as follows: First, the population aged less than 40 years was less likely to have LDCT records. Second, the data in the present study demonstrated that there was no significant difference in L1MI among ages by decades, even into the sixth decade. In other words, such extension would not influence the results if upper limit of the reference group was extended to 60 years. Additionally, if the reference group was defined as ages 20 to 60 years, it might more closely reflect the clinical use of LDCT in the real world.

The correlation between SMI and L1MI was relatively poorer in the reference group than in the COPD group (Appendix A). The correlation of SMI by BIA and cross-sectional muscle area by single slices of CT images is well documented [16,33]. Nonetheless, BIA may result in less precise estimates in situations in which the water-electrolyte balance is altered, such as dehydration, intake of alcohol and food, edema, and electrolyte imbalance [40]. Thereafter, instructions for accurate measurement of BIA composed of regulations of the aforementioned items and prohibition of exhausting exercise prior measurements [41]. However, about half of subjects with health checkup in the present study simultaneously fasted and took laxatives in preparation for gastroscopy and colonoscopy, which might disturb the fluid status and electrolytes balance. In addition, poor contact between the feet and electrodes may produce an error message. It is possible if the subject dose not recover fully from parenteral anesthesia for endoscopy examinations. These conditions may have caused the outliers of BIA measurement in the study population.

We analyzed the muscle surface in a single-slice CT at the L1 level using sliceOmatic software. Discrepancies in data between various software programs (including sliceOmatic and FatSeg, OsiriX, and ImageJ) are minimal and unlikely to be clinically significant [42,43]. Although we did not apply another software package for validation, the output of sliceOmatic segmentation software is reliable in L1MI measurement.

There were several limitations in this study. First, the individuals of the study received routine health check-ups at their own expense. Therefore, they were higher in social economic status and cared more about their health than the average population. Second, the LDCT in the study was predominantly applied in subjects aged less than 80 years. The application of the cutoff points to subjects older than 80 years might need to be studied further. Third, there were no well-established gold standards of determining muscle mass in the region of the study. It further limited the validation of the proposed cut-off points of L1MI.

In conclusion, the cross-sectional muscle area derived from LDCT in health check-ups is informative for evaluating muscle mass status. To our knowledge, this study is the first to establish the cutoff values of L1MI in Asia by examining more than 1000 healthy subjects. The cutoff value of L1MI was utilized in COPD patients. COPD patients with low L1MI might have more severe illness than those with normal L1MI.

## Figures and Tables

**Figure 1 jcm-11-02429-f001:**
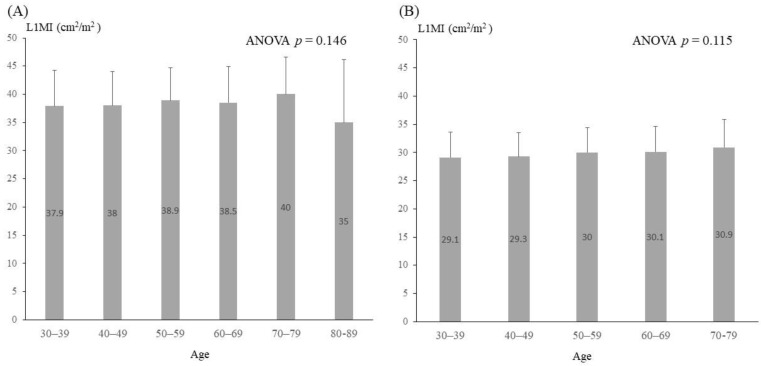
Histogram of mean L1MI by age decades in the study population. (**A**) Male (**B**) Female. Abbreviation: L1MI: the ratio of muscle surface (cm^2^) at L1 to height squared (m^2^). The numbers in the boxes are mean values. Vertical lines represent standard deviations.

**Figure 2 jcm-11-02429-f002:**
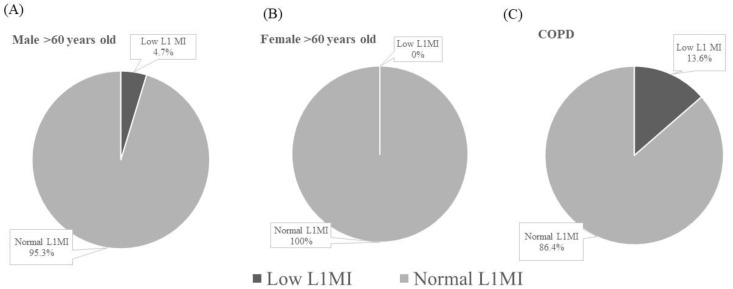
Proportion of low L1MI: (**A**) males aged more than 60 years; (**B**) females aged more than 60 years; (**C**) COPD. Abbreviation: L1MI: the ratio of muscle surface (cm^2^) at L1 to height (meters) squared; COPD: chronic obstructive pulmonary disease.

**Figure 3 jcm-11-02429-f003:**
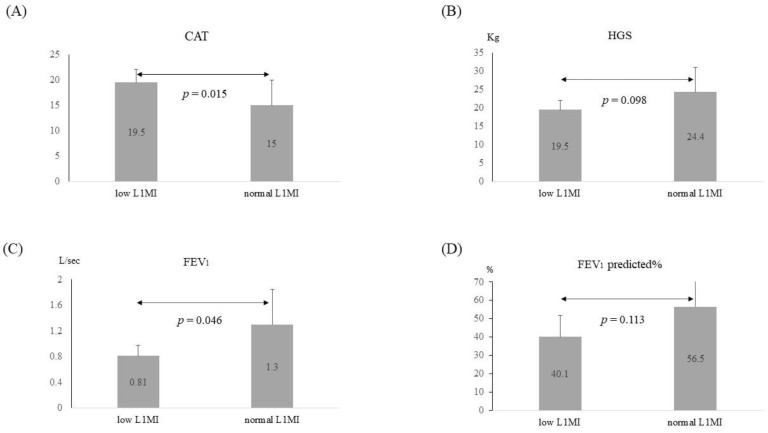
Comparison of CAT, HGS, FEV_1,_ and FEV_1_ predicted% between COPD subjects with low L1MI and normal L1MI. Comparing to COPD subjects with normal L1MI, those with low L1MI had (**A**) higher CAT scores (19.5 ± 2.6 vs. 15.0 ± 4.9), (**B**) marginally lower HGS (19.5 ± 2.6 vs. 24.4 ± 6.7 kg), (**C**) lower FEV_1_ (0.81 ± 0.17 vs. 1.30 ± 0.55 L/s), and (**D**) tended to have lower FEV_1_% of the predicted value (40.1 ± 11.8% vs. 56.5 ± 20.1%). Abbreviation: CAT: COPD assessment test; HGS: handgrip strength; FEV_1_: force expiratory volume in one second; FEV_1_ predicted%: the percentage of predicted FEV_1_; L1MI: the ratio of muscle surface (cm^2^) at L1 to height squared (m^2^); COPD: chronic obstructive pulmonary disease. The numbers in the boxes are mean values. Vertical lines represent standard deviations.

**Table 1 jcm-11-02429-t001:** The characteristics of the total population and sex-specific subgroups.

	Total (*n* = 1780)	Male (*n* = 1129)	Female (*n* = 651)
Age (years)	51.2 ± 11.1	51.4 ± 11.2	50.8 ± 11.0
20–29 *n* (%)	3 (0.2)	3 (0.3)	0 (0)
30–39 *n* (%)	330 (18.1)	202 (17.9)	120 (18.4)
40–49 *n*(%)	515 (28.3)	307 (27.2)	194 (29.8)
50–59 *n* (%)	579 (31.8)	362 (32.1)	205 (31.5)
60–69 *n* (%)	332 (18.2)	221 (19.6)	106 (16.3)
70–79 *n* (%)	57 (3.1)	31 (2.7)	25 (3.8)
≥80 *n* (%)	4 (0.2)	3 (0.3)	1 (0.2)
Height (cm)	165.6 ± 8.8	170.8 ± 6.4	157.9 ± 5.7 *
Weight (kg)	68.3 ± 13.5	74.5 ± 11.4	58.3 ± 10.2 *
BMI	24.7 ± 3.7	25.5 ± 3.4	23.4 ± 3.8 *
<18.5 *n* (%)	58 (3.2)	16 (1.4)	38 (5.8)
18.5–25 *n* (%)	975 (53.6)	520 (46.1)	435 (66.8)
25–30 *n* (%)	637 (35.0)	481 (42.6)	144 (22.1)
30–35 *n* (%)	130 (7.1)	101 (8.9)	28 (4.3)
>35 *n* (%)	20 (1.1)	11 (1.0)	6 (0.9)
SMI (kg/m^2^)	16.0 ± 5.7	17.3 ± 5.9	13.8 ± 4.7 *
L1MI (cm^2^/m^2^)	35.5 ± 7.2	38.4 ± 6.1	29.7 ± 4.4 *

Abbreviation: BMI: body mass index; L1: the first lumbar vertebra; L1MI: the ratio of muscle surface (cm^2^) at L1 to height (meters) squared; SMI: skeletal muscle index indicated the ratio of fat-free mass measured by bio-electric impedance analysis to height (meters) squared. * *p* < 0.001, males vs. females.

**Table 2 jcm-11-02429-t002:** The characteristics of gender-specific reference and older groups.

	Male		*p*	Female		*p*
	Reference Group (Age: 20–60) (*n* = 874)	Older Group (Age > 60) (*n* = 255)		Reference Group (Age: 20–60) (*n* = 519)	Older Group (Age > 60) (*n* = 132)	
Age (years)	47.1 ± 8.7	66.0 ± 4.5	<0.001	47.0 ± 8.6	65.9 ± 4.9	<0.001
Height (cm)	171.8 ± 6.3	167.2 ± 5.7	<0.001	158.8 ± 5.3	154.4 ± 5.6	<0.001
Weight (kg)	75.5 ± 11.7	70.8 ± 9.6	<0.001	58.2 ± 10.0	58.4 ± 10.9	0.990
BMI	25.5 ± 3.5	25.3 ± 3.1	0.348	23.1 ± 3.7	24.5 ± 4.0	0.001
<18.5 *n* (%)	11 (1.3)	5 (2.0)		32 (6.2)	6 (4.5)	
18.5–25 *n* (%)	403 (46.1)	117 (45.9)		365 (70.3)	70 (53.0)	
25–30 *n* (%)	363 (41.5)	118 (46.3)		98 (18.9)	46 (34.8)	
30–35 *n* (%)	87 (10.0)	14 (5.5)		19 (3.7)	9 (6.8)	
>35 *n* (%)	10 (1.1)	1 (0.4)		5 (1.0)	1 (0.8)	
SMI (kg^/^m^2^)	17.4 ± 5.9	17.1 ± 5.9	0.396	13.7 ± 4.6	13.9 ± 4.7	0.903
L1MI (cm^2^/m^2^)	38.3 ± 6.0	38.6 ± 6.5	0.510	29.6 ± 4.4	31.1 ± 6.2	0.133

Abbreviation: BMI: body mass index; L1: the first lumbar vertebra; L1MI: the ratio of muscle surface (cm^2^) at L1 to height (meters) squared; SMI: skeletal muscle index indicated the ratio of fat-free mass measured by bio-electric impedance analysis to height (meters) squared.

**Table 3 jcm-11-02429-t003:** The clinical characteristics of COPD patients with low and normal L1MI.

	Normal L1MI (*n* = 38)	Low L1MI (*n* = 6)
Male *n* (%)	33 (86.8)	5 (83.3)
Age (years)	74.4 ± 8.3	75.8 ± 5.0
Height (cm)	162.6 ± 8.3	160.8 ± 9.6
Weight (kg)	66.1 ± 11.7	49.1 ± 6.2 *
BMI	25.0 ± 4.2	19.0 ± 2.5 *
Hypertension *n* (%)	18 (47.4)	3 (50)
DM *n* (%)	7 (18.4)	2 (33.3)
Heart disease *n* (%)	15 (39.5)	2 (33.3)
CKD *n* (%)	5 (13.2)	0 (0)
CVA *n* (%)	3 (7.9)	0 (0)
Cancer *n* (%)	8 (21.1)	1 (16.7)
Cirrhosis *n* (%)	0 (0)	0 (0)
SMI (kg^/^m^2^)	18.2 ± 1.8	14.4 ± 1.2 *
L1MI (cm^2/^m^2^)	36.5 ± 5.9	25.4 ± 1.8 *
Frequent exacerbation	9 (23.7)	3 (50)

Abbreviation: BMI: body mass index; CAT: COPD assessment test; COPD: chronic obstructive pulmonary disease; CKD: chronic kidney disease; CVA: cerebrovascular disease; DM: diabetes mellitus; L1: the first lumbar vertebra; L1MI: the ratio of muscle surface (cm^2^) at L1 to height (meters) squared; SMI: skeletal muscle index indicated the ratio of fat-free mass measured by bio-electric impedance analysis to height (meters) squared. Frequent exacerbation was meant as emergency room visit more than once or ever admission history in previous one year. Continuous variables were tested by Mann–Whitney U test.; * *p* < 0.05, normal L1MI vs. low L1MI.

## Data Availability

The data that support the findings of this study are available from the corresponding author upon reasonable request.

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
