# Peer review of "Determination of Low Muscle Mass by Muscle Surface Index of the First Lumbar Vertebra Using Low-Dose Computed Tomography"

_jcm, 2022, doi:10.3390/jcm11092429_

Round 1
Reviewer 1 Report
Major comments
- A major concern is the use of ‘validation’ throughout the text. There are several types of validity, but none is made explicit in the text. It is important for the correct assessment of the applied methods (statistical ones included) and interpretation of the findings. In addition, no reporting guidelines are mentioned, which could improve the overall quality of the reporting.
- Lines 82-84. Additional details are needed regarding bioimpedance measurements. Full disclosure of the equipment, pre-measurement orientations, measurement protocol, and reference equations for the specific population should be provided as they can influence the validity of the results.
- Lines 110-112. I understand the concept of using multiples of SD as cutoff points, but I suspect reference 28 provided does not discuss this as a method. There are other methods for deriving cut-off points, such as receiver-operating characteristic (ROC) plots and related probabilities. It would be interesting to justify the selected approach, preferably with a proper (methodological) reference.
- Table 3. It is apparent the ‘Normal L1M1’ and ‘Low L1M1’ groups were categorized by the L1M1 value itself. Hence, the null hypothesis significance test for this variable between these groups is not needed as differences were set by such variable being a group-split criterion.
- Lines 311-315. Another major limitation is the lack of a gold-standard measure of muscle mass (which could make the adopted validity concept clearer for the reader). The lack of such measurement must be at least discussed as a limitation of the study.
- Figure S2 in the supplementary file. Inspection of the scatterplot for the reference group suggest at least two major trends. I would like to read some discussion about this.
Minor comments
- Lines 102-103. Please clarify whether the two pulmonologists and the radiologist assessments were independent of each other (e.g., the two pneumologists assessed the same images at the same time and provided a unique assessment, or two assessments were provided and combined by the radiologist).
- Line 151. The Pearson correlation coefficient is usually displayed as ‘r’ (lower case r).
- Some analyses are duplicated in tables and figure (e.g., table 3 and figure 3 both shows CAT, HGS, and FEV1). Please double-check if this is necessary for the interpretation of your findings.
Author Response
Comment # 1 A major concern is the use of ‘validation’ throughout the text. There are several types of validity, but none is made explicit in the text. It is important for the correct assessment of the applied methods (statistical ones included) and interpretation of the findings. In addition, no reporting guidelines are mentioned, which could improve the overall quality of the reporting.
Answer: We appreciated for your kindly reminding. The European Working
Group on Sarcopenia in Older People (EWGSOP) consensus recommends that cutoff points are set up at −2 standard deviations from the mean value of normative young and health reference population. There are several methods of measuring muscle mass proposed in EWGSOP2, including lumbar muscle cross-sectional area by computed tomography (CT). However, there are no universal gold standard about muscle quantities because ethnicity, regions and lifestyles might affect the determination of these cut-off points. Thereafter, we actually did not use the statistical validation method in the absence of the gold standard in our study. We would modify the term of validation in the manuscript.
Comment #2 Lines 82-84. Additional details are needed regarding bioimpedance measurements. Full disclosure of the equipment, pre-measurement orientations, measurement protocol, and reference equations for the specific population should be provided as they can influence the validity of the results.
Answer: Thanks a lot for your kindly reminding. TBF-410-GS, Tanita Inc, Tokyo, Japan was used in the study. TBF-410-GS is a tetrapolar foot-to-foot equipment delivering the frequency of 50 kHz and the current of 500 μA. TBF-410-GS had two prediction equations: standard set and athletics set. While importing the information of subjects, all measurements were preset as “standard set”. Then subjects will be asked to stand on the pressure contact stainless steel foot pads with bare feet independently. There were no specific pre-measurement orientation for subjects. The aforementioned information was supplemented in Appendix A.
Comment #3 Lines 110-112. I understand the concept of using multiples of SD as cutoff points, but I suspect reference 28 provided does not discuss this as a method. There are other methods for deriving cut-off points, such as receiver-operating characteristic (ROC) plots and related probabilities. It would be interesting to justify the selected approach, preferably with a proper (methodological) reference.
Answer: We thank the reviewer for the kind comments. As the aforementioned answer of comment #1, the cut-off points of muscle quantity might vary with ethnicity, regions and lifestyles. There is no well-established gold standards of low muscle mass in the region of the study. Besides, the aim of the study was to develop the cut-off points of L1MI by low dose chest CT, which there was sparse information in Asia. Therefore, we adopt the recommendation of EWGSOP about the determination of the cut-off points of muscle mass. We agreed that the reference 28 was not appropriately cited there. Alternatively, we cited EWGSOP as the reference.
Comment #4 Table 3. It is apparent the ‘Normal L1M1’ and ‘Low L1M1’ groups were categorized by the L1M1 value itself. Hence, the null hypothesis significance test for this variable between these groups is not needed as differences were set by such variable being a group-split criterion.
Answer: We thank for your kind comment. We agreed your comments that the null hypothesis significance test for this variable between normal and low L1MI groups was not necessary. Thereafter, we delete the presentations of p values in Table 3. However, we preserved the mark “*” which was representative of p< 0.05 in order that the readers could catch the information and the differences, while categorizing COPD patients by the proposed cut-off points of L1MI.
Comment #5 Lines 311-315. Another major limitation is the lack of a gold-standard measure of muscle mass (which could make the adopted validity concept clearer for the reader). The lack of such measurement must be at least discussed as a limitation of the study.
Answer: We are very appreciated for your kindly reminding. Indeed, there was one of the study limitations. We added the statement about the issue at Line 314-6.
Comment #6 Figure S2 in the supplementary file. Inspection of the scatterplot for the reference group suggest at least two major trends. I would like to read some discussion about this.
Answer: Thank you for the suggestion. BIA is an easy method to measure body muscle mass, compared to traditional dual energy X-ray absorptiometry, CT or MRI. The accuracy and reliability of BIA with foot-to-foot system mostly depends on the fluid status of the whole body and conductance of feet. Thereafter there might be some pitfalls about the measurement of BIA in health checkup. In order that the health checkup is completed efficiently in one day, panendoscopy and colonoscopy with parenteral anesthesia are performed at the same days. These subjects have to be fasting and colon preparation. Poor contact between the feet and electrodes may produce an error message. It is possible if the subject dose not recover fully from parenteral anesthesia for endoscopy examinations. The aforementioned issues might influence the reliability of BIA. We added the description at Lines 293-303.
Minor comments
Comment #1 Lines 102-103. Please clarify whether the two pulmonologists and the radiologist assessments were independent of each other (e.g., the two pneumologists assessed the same images at the same time and provided a unique assessment, or two assessments were provided and combined by the radiologist).
Answer: Thank you for giving our chance to clarify the point. We adopted that the images were independently assessed by two pulmonologists. The two assessments were confirmed by the radiologist finally. We added the statement at Lines 98-9
Comment #2 Line 151. The Pearson correlation coefficient is usually displayed as ‘r’ (lower case r).
Answer: We corrected it. Thank you
Comment #3 Some analyses are duplicated in tables and figure (e.g., table 3 and figure 3 both shows CAT, HGS, and FEV1). Please double-check if this is necessary for the interpretation of your findings.
Answer: We agreed on your opinions that CAT, HGS and FEV1 were duplicated in Table 3 and Figure 3. We deleted the data about CAT, HGS and FEV1 in table 3.
Reviewer 2 Report
This is a well-designed and comprehensive study that is highly relevant to the field of sarcopenia. Strengths of the study include its large study group size, as well as the establishment of L1M1 cutoff values followed by application of these values in COPD patients, all encompassed within the same study. The authors do address the major limitation that these cutoff values may be limited geographically/regionally to Asian populations. Nevertheless, given that low dose chest CT is being increasingly used as a lung cancer screening tool in other countries such as the US, this study sets a framework for establishing L1M1 cutoff values in relation to sarcopenia diagnosis based on low dose chest CT in other populations and geographic regions.
Overall, the study is well-written and clear in the description of its methods and results. One section of the discussion (Lines 268-274) does require editing to improve the clarity and accuracy of its message. The comparison between this study and the study by Derstine et al. here is confusing. I would focus on the differences in geographic region (US) and patient population (renal donor candidates) as accounting for the differences in cutoff values. Lines 268-270 suggest that the difference is due to Derstine et al. not “showing the differences of CAT scores and frequent exacerbations”, which is irrelevant as that study did not analyze COPD patients.
In addition, it is unclear how "ethics" apply to the comparison between the studies (Line 271), so that word may need to be removed. In lines 273-274, I do agree that the study is more applicable in the authors’ geographic region, but it is unclear how “differentiating the disease severity of COPD” is relevant to the cutoff values’ applicability in a particular geographic region.
Minor suggestions to fix errors and improve grammar and clarity are provided below.
—
Please remove “How to use this template” section at the beginning of the manuscript.
Line 72: space missing between “L1MI.” and “Therefore”
Line 222: remove extra period in “squared.;”
Line 239: consider changing “Thereafter” to “Therefore”
Line 251: consider changing “Kim and his colleagues” to “Kim and colleagues” or “Kim et al.”
Lines 252-253: for grammar and clarity, considering changing “derived from the data of abdominal CT collected in health check-ups” to “derived from abdominal CT data collected in health check-ups”
Lines 266-267: consider changing “reported by the study of Derstine and his colleagues” to “reported by Derstine and colleagues” or “reported by Derstine et al.”
Lines 284-285: consider changing “L1MI was reduced by the western lifestyle and convenient transportation in the younger population of today” to “L1M1 may be reduced by the western lifestyle and convenient transportation in the current younger population”
Line 286: consider changing “Thereafter” to “Therefore”
Line 291: change “as young reference group” to “as the young reference group” to maintain consistency with same phrase earlier in the paragraph
Supplementary File: The titles listed for the articles in the main manuscript vs. the supplementary file are not the same
Supplementary Table S1 title: consider changing “study of Derstine and his colleagues” to “study by Derstine and colleagues” or “study by Derstine et al.”
Author Response
Comment #1 Overall, the study is well-written and clear in the description of its methods and results. One section of the discussion (Lines 268-274) does require editing to improve the clarity and accuracy of its message. The comparison between this study and the study by Derstine et al. here is confusing. I would focus on the differences in geographic region (US) and patient population (renal donor candidates) as accounting for the differences in cutoff values. Lines 268-270 suggest that the difference is due to Derstine et al. not “showing the differences of CAT scores and frequent exacerbations”, which is irrelevant as that study did not analyze COPD patients.
Answer: Thanks a lot for your kindly comment. We revised the paragraph (Line 256-67) and focused on the possible etiologies that caused the differences of cut-off points. We agreed that the study of Derstine et al. was irrelevant to the application of COPD patients. We deleted the inference that the data of Derstine et al. applied to our COPD group.
Comment #2 It is unclear how "ethics" apply to the comparison between the studies (Line 271), so that word may need to be removed. In lines 273-274, I do agree that the study is more applicable in the authors’ geographic region, but it is unclear how “differentiating the disease severity of COPD” is relevant to the cutoff values’ applicability in a particular geographic region.
Answer: We were apologized that “ethnicity” was wrongly spelled as “ethics”. We corrected it. Besides, there was no gold standard of L1MI, which limited to verify the use of our proposed cut-off points. The relationships of COPD severity and sarcopenia were well-known. Thereafter, we applied the proposed cut-off points to COPD patients and investigate whether they could differentiate the disease severity of COPD or not. We added the prescription at Line 66-71.
Minor suggestions to fix errors and improve grammar and clarity are provided below.
Answer: All have been corrected, except that Supplements Table S1 was deleted.
Round 2
Reviewer 1 Report
Thank you for providing a response letter to my previous comments. All comments were addressed. I have no new comments.